# A Post-Colonial Ontology? Tim Winton's *The Riders* and the Challenge to White-Settler Identity

**Lyn McCredden** 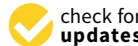

School of Communication and Creative Arts, Deakin University, Melbourne 3125, Australia;
lyn.mccredden@deakin.edu.au

**Abstract:** Through a reading of Australian non-Indigenous author Tim Winton and his novel *The Riders,* this essay seeks to shake to the very roots white-settler understandings of identity and belonging. The essay treads respectfully into the field of Australian identity, recognizing that Indigenous people's ancient and sacred relationship with country and the formation of treaties with the nation, are now rightfully central on national agendas. However, this essay asks the following question: what are the ontological grounds upon which respectful *dialogue* between Indigenous and non-Indigenous Australians might occur, after such violent and traumatic history? The essay explores the possible grounds for an evolving dialogue, one which will be necessarily intersectional: (post)colonial, spiritual/ontological and material. Further, the essay identifies "spirituality" and "ontology" as broad denominators for religion, speculating on a (post)colonial ontology which centers on home and (un)belonging. White-settler Australians, this essay argues, must confront deep ontological issues of brokenness if they are to take part meaningfully in future dialogues. Scully, the protagonist of *The Riders*, finds himself far from home and stripped of almost all the markers of his former identity: as Australian, as husband, and as a man in control of his life. The novel probes (un)belonging for this individual descendent of colonial Australia, as trauma engulfs him. Further, it will be argued that *The Riders* prefigures the wider, potentially *positive* aspects of a post-colonial ontology of (un)belonging, as white-settler Australians come to enunciate a broken history, and ontological instability. Such recognition, this essay argues, is a preliminary step towards a fuller post-colonial dialogue in Australia.

**Keywords:** post-colonial; white-settler identity; spiritual; Tim Winton; ontology; Belonging and (Un)belonging

## 1. Introduction

*The Riders*, and indeed the entire fiction oeuvre of Australian novelist Tim Winton, has garnered more than its share of controversy. Winton is sometimes dismissed as writing only from "white-settler" perspectives at a time when Indigenous Australia is urgently seeking recognition of its ancient peoples. Some critical accounts see Winton as "too Christian" (see Goldsworthy 2009), a non-Indigenous author at a time when white Australia should perhaps be listening to Indigenous voices (see Griffiths 2014). Significantly, *The Riders* is the only Winton novel set outside Australia, in a cold, wintry Irish landscape that is alien, but uncannily connected to Scully, the white Australian protagonist. With this setting, Winton is able to explore questions of dispossession and migrancy in ways that echo, but are necessarily very different to, Indigenous concerns about dispossession and belonging. So what does *The Riders* have, speaking strongly from a white-settler and individualizing set of perspectives, to contribute to any kind of understanding of a post-colonial ontology?

*The Riders* has principally been read for its traumatic personal narrative; but this essay seeks to draw attention to the wider post-colonial and spiritual/ontological landscapes in which the main character,

Scully, has been formed. Scully is a homebody, an Australian man drawn reluctantly away from home in search of his wife, who has intentionally disappeared, leaving her child and husband abandoned and traumatized. As the novel develops, the volatile processes of a distinctively white-settler ontology take on not only personal but wider spiritual significance. All of Winton's novels, from the early works such as *An Open Swimmer* (Winton 1982) to his most famous and loved novel, *Cloudstreet* (Winton 1991), as well as later novels such as *Dirt Music* (Winton 2001), *Breath* (Winton 2008), *Eyrie* (Winton 2013) and *The Shepherd's Hut* (Winton 2018), are set locally in Australian, indeed Western Australian, landscapes. This essay will argue that in Scully's displacement to Ireland and across many scenes in Europe, a fixed notion of identity (of author and characters) is being unpicked, as we are presented with a white Australian man in free-fall, stunned by his own lack of power, subject of vertigo and loss. The narrative becomes not merely that of an individual but of necessary white-settler reversals, creating the grounds for a new, less complacent, less settled ontology. However, it is important to say at the outset that there is no intention in this essay to equate Indigenous displacement and loss of "home" with one white man's trauma. This would be totally disproportionate. What this speculation on a possible white-settler ontology seeks to do, in tracing the losses and displacement of Winton's protagonist, is to ask what it takes for non-Indigenous Australians to be able to listen and to comprehend the depths of violent displacement and ontological vertigo. Instead of firm, complacent Australian identity, Scully grasps at the disappearing shreds of his once-comfortable, seemingly settled Australian world.

Beyond the individual character, this essay will argue that a new ontology of unbelonging is what is needed, as white-settler Australia acknowledges the brokenness of its history. New dialogues might become possible between Indigenous and non-Indigenous Australians as the blindnesses of a white-beach, unquestioning white-settler Australia are addressed.

For Indigenous activist and critic Aileen Moreton-Robinson, white Australia needs to understand its own history of migrancy. Moreton-Robinson argues that there is of course no single Australian ontology but an ongoing series of lived and debated conditions, with very different parameters for Indigenous and non-Indigenous Australians and differing definitions of the ontological realities of these groups, even *within* each group. In her enormously influential 2015 critical volume *The White Possessive*, Moreton-Robinson writes definitively of these differences, arguing:

> In postcolonizing settler societies, Indigenous people cannot forget the nature of migrancy, and we position all non-Indigenous people as migrants and diasporic. Our ontological relationship to land, the ways that country is constitutive of us, and therefore the inalienable nature of our relation to land, marks a radical, indeed incommensurable, difference between us and the non-Indigenous. This ontological relation to land constitutes a subject position that we do not share, that cannot be shared, with the post-colonial subject, whose sense of belonging in this place is tied to migrancy. (Moreton-Robinson 2015)

Moreton-Robinson writes powerfully and passionately from an Indigenous perspective about the need for non-Indigenous Australians to acknowledge their migrancy. She does this against what she sees as "a radical, indeed incommensurable, difference between us and the non-Indigenous." Such differences cannot be addressed in this essay; nor can her firm view that Indigenous' "ontological relation to land constitutes a subject position that we do not share." While these are important contentions which need to be part of any ongoing dialogue, what is core to the arguments in this essay about *The Riders* is her tying together of white-settler Australia with migrancy. To belong, to have a home, to experience a relation to land, are all ontological markers long recognized by Indigenous Australians. Undoubtedly, these ontological markers need to be more fully explored by white-settler Australians, descendants of those who violently possessed the land in which they now live.

One outcome of such questioning, over the last three decades, has been the wealth of fiction and criticism and its probing of identity in diverse post-colonial nations: from Stuart Hall's seminal 1990s work on Caribbean and new British identities (Hall 1990); to Bhabha (1990) and Spivak (1990) on Indian diaspora; to Indigenous and non-Indigenous selfhood in the critical writing of Muecke

(1997, 2004), Rose (2004, 2011) and Gibson (2012) in Australia. The fiction and poetry of many Australian Indigenous, diasporic and post-colonial writers, which includes the work of Narogin (1991), Wright (1997, 2006), Scott (2010, 2017), Birch (2019), Cobby Eckermann (2015) and Harkin (2015), amongst many Indigenous peoples globally, has been confronting and destabilizing old ontological sureties (white, male, Western, imperial). *The Riders* too asks complex and uncomfortable questions about post-colonial identity, and concomitantly of white-settler European desires for belonging in Australia. Through a reading of the novel this essay will speculate on what is arguably the next step in Indigenous/non-Indigenous dialogues. This "next step", as with the term "dialogue", is indeed speculative. It premises a new ontological understanding of white-settler Australian identity: a new ontology both spiritual and material, requiring non-Indigenous Australians' acknowledgement of their brokenness and the "migrancy" of their (un)belonging.

## 2. Identity: Imperial Process, Fictive Re-Imagining

Identity is *agonistically* represented in *The Riders*. That is, rather than offering politically acceptable or already-resolved, politically unrealistic depictions of "Australian" identity, often in static binary terms (colonial/post-colonial, sacred/secular, them/us), Winton's fiction can be read as wrestling, *in medias res,* with the intersectional and fluid parameters of identity-making, and the ways these identities are still being negotiated and re-imagined in Australia.

Diana Fuss has written on the inheritance of psychoanalyst and early theorist of race Franz Fanon, arguing "that the psychical operates precisely as a political formation . . . a politics that does not oppose the psychical but fundamentally presupposes it." (Fuss 1994, p. 39). Fuss is primarily focused on *racial* identification, and what is suggestive for this current essay is that if we accept Fuss's premise, both psychic (ontological, spiritual, ego-forming) and political (post-colonial, national, racial) aspects of identity-formation are inextricably bound together for all subjects formed in the crucible of colonial rule. Fuss, in Franz Fanon's footsteps, offers an historically nuanced account of simultaneously psychoanalytic and post-colonial approaches to identity formation:

> The colonial-imperial register of self-other relations is particularly striking in Freud's work, where the psychoanalytic formulation of identification can be seen to locate at the very level of the unconscious the imperialist act of assimilation that drives Europe's voracious colonialist appetite. Identification, in other words, is itself an imperial process, a form of violent appropriation in which the other is deposed and assimilated into the lordly domain of Self. Through a psychical process of colonization, the imperial subject builds an Empire of the same and installs at its centre a tyrannical dictator, "His Majesty the Ego." (Fuss 1994, p. 23)

What Fuss's analysis suggests is that for any critical consideration of post-colonial fiction, there needs to be reflection on the different ways in which characters (those from colonized countries, whether Indigenous, white settler or multicultural/diasporic) encounter, and are formed by, "Europe's voracious colonialist appetite". This is equally true for characters who resist or embody such voraciousness. In *The Riders*, Scully is an Australian male character of white, European heritage. His journey from Australia to Europe is not a return to the "homeland", as he considers himself a fully Australian, indeed Western Australian man from white beaches and sunny climes. While there is no towering "lordly domain of Self" being wielded by Scully in his seeking of identity and home, his complacent insularity before his journey can be read as needing to be unfixed from its sense of complacent, blinkered Australian belonging. Scully is hardly a "tyrannical dictator" in Fuss' terms, no heroized discoverer of "His majesty the Ego." Rather, he is the chaotic subject of traumatic loss—of homeland, wife and former identity. Against the character's cosy sense of familiarity with "home" and all its comforts, Winton places multiplicity and the necessary disruption of identity in Scully's ontological journey. Moved from a place of familiarity in his Australian beach suburb, to one in which he is ill at heart, displaced, Scully begins to suffer vertigo of identity.

Many of Winton's novels radically destabilize the identity of male white-settler characters—Luther Fox in *Dirt Music*, Bruce Pike in *Breath*, Tom Keely in *Eyrie,* and Scully, the deserted husband, undermined in his gendered confidence, longing for his lost wife. It is in the *agonistic and realistic* representations of white-settler identity that Tim Winton's fiction is significant, persisting as it does across his oeuvre in asking awkward questions about still-evolving post-colonial identity-making, and the ontological depth of non-Indigenous brokenness and (un)belonging.

## 3. The Unmanning of Scully

Significantly, *The Riders* depicts Scully's sense of unbelonging in Ireland, the place to which he has travelled far from home, mainly at the instigation of his wife. Ireland, itself a deeply migratory, colonized nation, and the "homeland" of many white-settler Australian descendants, is the perfect setting for exploration of such migrancy. Scully is homesick, unused to the cold Irish landscape he arrives in, and working blind to build a new home for his family from the rubble of an old building. As the terror of what is happening to him first begins to dawn on Scully, that his wife has disappeared, and probably through choice, Winton writes of Scully's vertigo, and the vulnerability of his identity: unanchored, no longer husband, no longer belonging in Australia. Through a series of objective correlatives that powerfully mimic the frozen, uncomprehending emotions of the man, his inability to read his situation by any of his known parameters, the narrative deftly dramatizes the sense of terror and displacement that is creeping over Scully:

> The sun was gone before four o'clock. Scully found himself out behind the barn in a strange cold stillness looking at the great pile of refuse he'd hauled out there on his first day . . . a slag heap, a formless blotch here at his feet. In the spring, he decided, he'd dig up this bit of ground and plant leeks and cabbage, and make something of it. Oh, there were things to be done, alright. He just had to get through tonight and the rest of his life would proceed. (Winton 1994, p. 98)

What does such an externalized description of inner turmoil evoke in readers? We might empathize with Scully's need to mend, build and control, in the face of the powerlessness and vulnerability that is coming over him. Scully seeks to defend himself against the brokenness of his life, to reassert his known boundaries, his identity, to do what is familiar; but what is also evident is that he is an ingénue, inadequate or uncomprehending in the face of the grief that is pouring over him like the "strange, cold stillness". The need to build, grow, provide, "make something of it", "to get through tonight", creates an ontological state of terror, but also a need to overcome, to reassert. We can think through Scully's character in a number of frames: colonial, gendered, ontological, all at the very edge of self-containment.

Such a fragmenting sense of identity is described in Judith Butler's 2004 critical volume *Precarious Lives,* as she analyses the ways in which loss undoes human borders, producing terror. But Butler also explores a productive openness, or at least a humility borne of brokenness, that might arise from experiencing such loss. In this latter frame, Scully can be seen on a similar quest to that of Jennifer, his wife. Both are facing displacement in their lives. This is Butler's notion of the unfixing of identity, akin to Julia Kristeva's state of abjection, an undoing that comes primarily from the shock of loss, producing as it does in *The Riders,* a fracturing of identity. For Jennifer, this seems to have been desired and chosen; for Scully it is imposed as trauma. Both are personal losses, but also trans-personal in their recognizable, human dismantling of identity.

The pain of seeking, or being forced, to emerge into mobile, less-fixed identity, is a central concern of many post-colonial novels which trace national and personal identity-making in places always already scarred, a process mightily magnified for those of displaced societies, such as Indigenous and diasporic peoples, who have experienced displacement for centuries. The sometimes brutal psychic consequences of such loss, and an emergence into a new ontological awareness, can be seen in *The Riders* in both intimate and broader terms through the mechanisms of readerly empathy. As Scully

roams far from what he remembers, the known comforts of his Australian landscape, and without his wife, identity cannot be simply and finally resolved, parceled out into discrete modes of ontology which inevitably lead to a sense of belonging. (Un)belonging becomes Scully's new ontological reality. As Butler writes:

> When we lose some of these ties by which we are constituted, we do not know who we are or what to do. On one level, I think I have lost "you" only to discover that "I" have gone missing as well. At another level, perhaps what I have lost "in" you, that for which I have no ready vocabulary, is a relationality that is neither merely myself nor you, but the tie by which those terms are differentiated and related. (Butler 2004, p. 12)

This loss of the other, and accompanying loss of self, powerfully illuminates the core understandings of identity in *The Riders,* but also throws a scouring light back across loss of self, family and community for Indigenous Australians. While Butler's frame is primarily personal and intimate, many readers of such *personal* ontological loss may also think of the slowness of white Australia to understand the impact of such overwhelming loss. In *The Riders,* this destabilizing of seemingly robust Australianness in Scully results in his mad and nightmarish quest across Europe. The multiple experiences of loss are seen to undo the identity of the one left behind, the one without anchor, and to plunge them into a grief in which both the one who is gone and the one who is in mourning are understood to lose selfhood, to "go missing". Scully is made to step outside himself, to reach through otherness of many kinds, in his enforced quest.

The radical tension between being at-home, comfortable and known-in-place, and a growing awareness of embodying, being the agent of, colonial and imperial destruction, is, for many white-settler Australians, a very real and present ontological dilemma. If we read *The Riders* in this context, Winton is asking a deeply ontological question: what happens to white-settler Australians when such certainties of identity are powerfully unfixed?

If we think about Butler's main contentions in relation to *The Riders*, we might want to pursue a more troubling thread of the struggle for identity taking place between self and other and about the parameters of ontological identity formation in colonized societies. Both Jennifer and Scully are travelling away from any fixed or safe understanding of self—in place, in national identity, in relationship to otherness—whether they want to or not. Jennifer is portrayed through silence and absence, but what we know is that she has taken powerful steps to liberate herself from the known in her life, from motherhood, marriage, homeland, intimacy with Scully. Rather than simply judging her, the novel's proposal is to read her, and Scully, through the dynamics of absence, of what they once had, glimpsed through Scully's grief and his half-demented journey across Europe. As Scully reels from the events unfolding for him, he is both grieving his past and speaking into the void of the future, without intimate relations, without homeland. He is the white, privileged, male self, unanchored. As Butler puts it, the story we tell in the midst of grief is wild, abandoned and inchoate, which seems an accurate description of Scully's narration. Butler writes further:

> I might try to tell a story here, about what I am feeling, but it would have to be a story in which the very "I" who seeks to tell the story is stopped in the midst of the telling; the very "I" is called into question by its relation to the Other, a relation that does not precisely reduce me to speechlessness, but does nevertheless clutter my speech with signs of its undoing. I tell a story about the relations I choose, only to expose, somewhere along the way, the way I am gripped and undone by these very relations. My narrative falters, as it must.

> Let's face it. We're undone by each other. And if we're not, we're missing something.

> This seems so clearly the case with grief, but it can be so only because it was already the case with desire. (p. 13)

The faltering narratives of post-colonial Australian identity have formed Scully and Jennifer from the beginning of their relationship, unmanning him and shaping her narrative in ways the novel only hints at. Scully had wanted to remain where he was, on his beloved beaches, amongst family and friends, in a known landscape which he thought he possessed. Jennifer, for reasons only guessed at by Scully, wanted to move away, to be somewhere else. The tension between "home" and "elsewhere" is well known to many colonial subjects, though radically differently for Indigenous and non-Indigenous.

## 4. Irish Ruins and Wounded Warriors

Scully's journey takes on further levels of vertigo as he walks one day into the Irish landscape, drawing his personal losses with him through the Norman castle ruins and the "rubble-strewn pit of the great hall", where "[e]verything had fallen through onto everything else. Great oak beams lay like fallen masts and rigging across cattle bones and tons of cellar bricks". (pp. 49–50) This is a European landscape alien to Scully. This is another kind of colonized landscape—post-feudal, post-imperial. The novel here presages the abject journey into Europe that Scully is about to undertake, that he is personally impelled towards: a kind of reverse or impotent colonialism, a stark revealing of the unhomely to him. It is also an Australian's reading of Europe: discomforted, foreign, not-home. Hence, Scully's first, mysterious and ghostly encounter with the eponymous riders comes at a moment of rising personal anxiety, the night before he is hoping to reunite with Jennifer and Billie:

> Scully moved between the riders, all but touching the heaving, rancid flanks of their mounts. Some of the horses had black, congealed wounds on their chests, and they looked as tired and cold and dazed as their riders. Some were boys, their scrawny legs bare and stippled with gooseflesh. And how they craned their necks, these riders. It was as though any moment some great and terrible event would explode upon them, as if something, someone up there could set them in motion . . . He felt himself craning, waiting, almost failing to breathe . . . His feet took root in the ground as they continued to wait and he waited with them. It was true, he knew it, something was about to happen.

> But the awful stillness went on. (p. 81)

Little has been written about the symbolic suggestiveness of the riders, these figures who appear in the night, in Scully's moments of wildest anxiety. They are figures of the ancient European past, warriors coming from savage scenes of battle, wounded boys and men, haunting presences, deeply un-Australian in their armor and chainmail; what they portend is profoundly ambiguous. Are they to be read in alignment with Scully's mental state, as mirrors of his own losses; or are they the embodiment of all the *otherness* Scully is being required to confront, as self and other warp and waver in fearful recognition? This movement between self and other is highly significant for the novel's ontological insight.

As so often in *The Riders*, the figures act as objective correlatives of Scully's internal terror; but they are also European warrior figures from a world and a history that Scully doesn't know. They are both threatened themselves and terrifying to Scully. This ambiguity in what they signify is partly because their wild, wounded appearance, from a world beyond Scully's known, supposedly settled, Australian sense of safety, rears up in a landscape which is not his own; and yet there is some form of recognition of them by Scully. They are also mirrors of shared trauma and loss, embodying the psychic terror, the rushing, oncoming events of life beyond control; men, and boys, savaged by the violence of their initiation into the traumas of some imperial war or other. It is the dramatization of an identity in psychic free-fall that the scene reflects as Scully and the warriors confront each other.

Anthropologist Greenwood (2008) has written of the pan-European myth of "The Wild Hunt", to which the riders in this novel seem affiliated. Amongst the related myths and heroes of the Wild Hunt, led sometimes by men and sometimes by women in different narratives, are the Germanic god of the wind and the dead, Wodan, the goddess Freya, and the Irish mythical hunter Fionn mac Cumhaill, or Finn MacCool, whose son was Oisin. Greenwood writes:

> As far as practitioners of nature spiritualities are concerned, the Wild Hunt offers an initiation into the wild and an opening up of the senses; a sense of dissolution of self in confrontation with fear and death, an exposure to a 'whirlwind pulse that runs through life'. In short, engagement with the Hunt is a bid to restore a reciprocity and harmony between humans and nature. (p. 220)

A wild hunt is what Scully has embarked on, though he doesn't rationally know it yet. *The Riders* both broadens and intensifies the significance of the wild hunt motif, expanding and destabilizing the parameters of Scully's Australian identity. The terror the riders draw out in Scully, "a sense of dissolution of self in confrontation with fear and death", is close to the kind of undoing of self which Butler writes about, when the self is "periodically undone and open to becoming unbounded". (Greenwood 2008, p. 220). The novel thus depicts the eruption of the wild, dark, historical European world into the life of a supposedly post-colonial Australian man who had believed himself originating from untrammeled horizons and easy belonging. His homesickness and his psychic displacement in Europe see him whirling through a hyperreal landscape, seeking her, the embodiment of all he thought gave him meaning. But home and a sense of belonging and intimate acceptance prove to be less and less attainable. This privileged white Australian man is made to confront ontological brokenness and unbelonging.

As Scully journeys, unable to grasp what is happening, what he wants or where he should be, his narrative becomes increasingly disoriented, haunted, faltering. "Of course she left you—there's nothing to you", (p. 167) he concludes during his drunken night-time swim in Hydra. In the early encounter with the riders, in the European journey, and in the final scene of the novel, Scully's ego is opened out to threat, change, and unknowing. In the final moments of the novel he is depicted as both one of the riders, and yet not so:

> With his wild hair and arms, his big eyes streaming in the firelight turned up like theirs to the empty windows of the castle, he was almost one of them . . . but he knew that as surely as he felt Billie tugging on him, curling her fingers in his and pulling him easily away, that he would not be among them and must never be, in life or death. (p. 377)

The novel here makes its complex resolution, as Scully turns away from the dark abjection of the riders and necessarily begins constructing an ego boundary, a survival tactic that returns him to his daughter and the future. He is saved by his child, which is a common Winton gesture. However, the more unnerving insight of this dénouement is to be found in Scully's act of will against what "must never be", as he erects boundaries. Yet even as he reasserts his ego, there is also a sense of *affiliation with* the riders: "He recognized the blood and shit and sweat and fear of them, and he looked into the dead heart of the castle keep . . . whose light did not show and whose answer did not come." (p. 377). Scully is here glimpsing the deep ambiguity and hauntedness at the heart of post-colonial ontology. He is a man asserting an act of will, the ego as surviving through the necessary setting of boundaries; but boundaries are recognized simultaneously as temporary and precarious. It is a doubly haunted ego inhabiting Scully. Behind this personal narrative stand the figures of white-settler Australia, some of whom are the heirs of those wild-haired, displaced warriors, Irish and imperial, victims and perpetrators, both ghosts and very real, in the ongoing narratives of post-colonial Australia. Again, the relations between self and other are tested, stretched almost to breaking, but hold finally in place.

As Butler describes the formation of the self in trauma:

> I may wish to reconstitute my "self" as if it were there all along, a tacit ego with acumen from the start; but to do so would be to deny the various forms of rapture and subjection that formed the condition of my emergence as an individuated being and that continue to haunt my adult sense of self with whatever anxiety and longing I may now feel. (Butler 2004, p. 16)

Butler's insights are pertinent in considering Winton's representation of individual psychic needs but also equally in their broader, ontological implications. Threading their way through so many

other contemporary Australian novels by white-settler authors—such as Kate Grenville's *The Secret River* (Grenville 2005), Andrew McGahan's *The White Earth* (McGahan 2004) and Gail Jones' *Sorry* (Jones 2007)—as well as, differently, through novels by Indigenous authors—such as Alexis Wright's *Plains of Promise* (Wright 1997), *Carpentaria* (Wright 2006) and Kim Scott's *That Deadman Dance* (Scott 2010)—dramas of traumatic post-colonial ontology can be traced. This ontology is a conjunction of personal and communal desires for belonging in the midst of deep historical losses. While individual selves must, finally, revert to the boundaries of recognizable identity in order to go on living, this novel asks if is there is the possibility of glimpsing the needs of the other, of acknowledging the other's homeland, through the processes of loss and brokenness.

In Winton's *The Riders,* the continuous formation and deformation of the self in relation to the other is posited at the intersection of ontological and post-colonial processes of identity-formation. As this essay has been arguing, at their finest, post-colonial novelists and critics work in the name of a belonging always aware of its unbelonging. This is, as this essay has been arguing, a white-settler ontological condition. But for many Indigenous, as well as white-settler Australians in different ways, belonging and the forging of identity have been undertaken through loss and trauma. For Butler, this means a struggle that never ceases, the self as a tentative telos, dwelling in a belonging which must always be tempered with a realization of unbelonging. This can be understood as an ontological and spiritual stance as much as it is a post-colonial condition. The author imagines the struggles of his characters as they come towards understanding of this condition. Whether in Australia or in other places, they are strangers who must confront loss of hegemonic identity if there is to be any future possibilities, any new dialogues. Winton is, if this essay's premise is to be accepted, a novelist of white-settler Australia whose probing of ontology does not privilege white Australian belonging but seeks to shake to its very roots any idea of an imperial, privileged, complacent identity.

**Funding:** This research received no external funding.

**Conflicts of Interest:** The author declares no conflict of interest.

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
