# Peer review of "A Post-Colonial Ontology? Tim Winton’s The Riders and the Challenge to White-Settler Identity"

_humanities, doi:10.3390/h9030095_

Round 1
Reviewer 1 Report
The essay has potential but it requires more effort. The argumentation is not clear and goes in many directions. The author aims to analyze a multiplicity of issues in Tim Winton's "The Riders", such as spiritual and postcolonial struggles confronting white settlers in Australia, new ontological and spiritual ontologies, and Winton's representation of gender. The theoretical introduction is rather too long and moves in a completely different direction, with several references to Butler's "Precarious Lives". The connection between this theoretical opening and the following interpretation is unclear.
If the author wishes to analyze postcolonial trauma (which is not clearly stated as their intention), references to postcolonial trauma theory would be more relevant here (works by Stef Craps in particular). Moreover, the spiritual aspect of the novel is approached in a shallow way. The connection with gender is not really developed either. What is the postcolonial ontology proposed by Winton remains unclear.
It would be advisable for the author to limit the scope of reflection and to elaborate, for instance, an in-depth analysis of spirituality in the novel. This would require a different analytical framework, though.
Punctuation to be corrected. Some sentences are very long, the style sometimes rather informal.
Author Response
Assessor 1
Thank you so much for your careful reading of my essay. In response to you I have:
1.written a new abstract and introduction, which I think streamlines and clarifies the argument of the essay;
- cut down considerably on the long theoretical section;
- deleted the gender concerns, in order to streamline the argument around ontology/spirituality and post-colonialism;
- offered stronger definition and content throughout to what is meant by new ontological/spiritual insights of the novel. Hence, there's a stress now on the brokenness of the protagonist, in parallel with the brokenness necessary for post-colonial Australians to acknowledge, if there is to be a more fruitful dialogue around Australia's violent history;
- Butler has been used more sparingly, and better links made between the novel's individual in trauma and a wider, post-colonial understanding of trauma between Indigenous and non-Indigenous Australians. I have tried to make it very clear that the essay, and the novel, are concerned with white-settler Australians, who need to open up ontologically to the other, and to new possible dialogues.
- a new section quoting Indigenous critic Aileen Moreton-Robertson has been included at the beginning of the essay;
- Sentence lengths have been curtailed throughout.
Reviewer 2 Report
The essay does the best it can to dislodge Winton from a mainstream white Australian identity and aligning his work with the sport of postcolonial and antiracist theory and practice with which it has rarely been aligned, I agree both that there is much with the work of Tim Winton, counter to his mainstream appropriation by the Australian print culture and education systems, to justify this and that the essay does an effective job in making the argument. I am not sure The Riders is the strongest book to make this argument, because of its largely non-Australian setting that it’s what the author has chosen for that example and that is the ground upon which we have to work. But I remain unsure whether a lot of Scully’s anxieties are the same as that of any existentially isolated subject without recourse to settler identity, and whether evne in that vein Winton was not repeating in The Riders what had already bene done by American authors such as Paul Theroux in the Mosquito Coast.
But, without simply thwacking the white male author for being white or in a reverse-racist way privileging the indigenous or non-white subject over the white subject, I have to wonder if the dynamics of Winton’s agenda precisely dovetail with the post-colonial, anti-imperial and reconciliations gender outlined there. I hasten to add that I think there can be distinction here without utter difference, and that Winton’s road and the roads of the indigenous authors mentioned in the essay might represent different journeys to the same goal. I cannot see themas tantamount to each other, however.
A lot of the theoretical essays cited (Butler, Kristeva, Povinelli) are a bit older in terms of the date of their publication all being fifteen years or so ago, and I am surprised not to see more recent work by Veracini for instance in the bibliography. In terms of the criticism on Winton himself the author is quite up to date. Citing some relatively recent Australian theory by Alison Ravenscroft, Jennifer Rutherford, Michèle Grossman, and Anita Heiss might help his out a bit. It is by the way Deborah Bird Rose, not Debra.
Author Response
Thank you so much for carefully reading my essay. Please note that a new abstract has been written, in line with the major changes to the manuscript. In response to your assessment, I have:
- offered a fuller explanation as to why "The Riders", being set away from Australia, an unusual occurrence in Winton, is a highly appropriate work with which to consider the gist of post-colonial ontology, as entailing "unbelonging" and displacement;
- tried to convince you of the ways the individualist approach of the novel, centering on Scully's trauma, is paralleled with the wider, post-colonial ramifications of displacement, fragmented identity, around white-settler identity;
- agreed totally and sought to give further content to your sentence "...Winton’s road and the roads of the indigenous authors mentioned in the essay might represent different journeys to the same goal." throughout the essay;
- renovated the theoretical and critical references in the essay, keeping Butler (though a little cut down),and Fuss, and added some of the important work of Indigenous critic Aileen Moreton Robinson at the beginning of the essay;
- omitted the gender references throughout the essay, as I think the essay was becoming too multi-stranded;
- offered stronger definition and content throughout to what is meant by new ontological/spiritual insights of the novel. Hence, there's a stress on the brokenness of the protagonist, in parallel with the brokenness necessary for post-colonial Australians to acknowledge, if there is to be a more fruitful dialogue around Australia's violent history.
Thanks once again.

Round 2
Reviewer 1 Report
The manuscript has been substantially improved and can now be recommended for publication. The author has clarified her/his argumentation, with more focus on post-colonial identity and spirituality. The essay is more logical and coherent now. Though the proposed reading of Butler is controversial (I would think of the cited work as relating mostly to loss and grief), it can also be considered innovative, as applying to Australian identity. The reference to Moreton-Robinson has definitely clarified the author's point.
In the revised version there is an excessive use of the word post-colonial - I would kindly ask the author to reconsider this. I would also suggest revising the last sentence - I am not sure "The Riders" really offers such a radical questioning of traditional approaches to identity.
line 355: Butler does not write about migrancy: these are your conclusions
Author Response
Thank you once again for your attentiveness to my essay. I have made a number of changes in line with your recommendations.
I have deleted the term "post-colonial" at the following lines: 22, 44, 159-60, 267, 276, 278 (and added "of race"), 295, 809, 1044 (replaced with "colonized societies"), 1053, 1070 (replaced with "colonized"), 1332, 1376, 1380 (added "supposedly"), 1551 (and added "privileged white"), 1573, 1585, 1598, 1789 (and added "This can be understood as an ontological...").
I have deleted "migrancy" in relation to Butler, as recommended.
I wish to stay with my final sentence, as it is in part a defense of Winton's contribution to post-colonial issues, which has been underappreciated/misunderstood in Australia particularly.
I am attaching the final, amended manuscript below.
Thank you once again for your attention to my essay.
